# Quantifying Likeness: A Simple Machine Learning Approach to Identifying Copyright Infringement in (AI-Generated) Artwork

## Abstract

Through study of legal precedent, we propose a pragmatic way to quantify copyright infringement, via stylistic similarity, in AI-generated artwork. Copyright infringement by AI systems is a topic of rapidly-increasing importance as generative AI becomes more widespread and commercial. In contrast to typical work in this field, more in line with a realistic legal setting, our approach quantifies similarity of a set of potentially-infringing "defendant" artworks to a set of copyrighted "plaintiff" artworks. We develop our approach by making use of one of the most litigated artistic creations of this century – Mickey Mouse. We curate a dataset using Mickey as the plaintiff, and perform hyperparameter search, scaling, and robustness analyses with various defendent artworks from real legal cases to find settings that generalize well. We operationalize similarity via a simple discrimintative task which can be accomplished in a low-resource setting by non-experts – our aim is to provide a 'plug and play' method that is feasible for artists and/or legal experts to use with their own plaintiff sets of artworks. We further demonstrate the viability of our approach by quantifying similarity in a second curated dataset of Maria Prymachenko's art vs. AI-generated images. We conclude by discussing uses of our work in both legal and other settings, including provision of artist compensation.

## 1 Introduction

The rapid development and widespread use of generative AI models has raised concerns among creators about potential job disruption and copyright infringement. In the courts, these technologies are challenging our understanding of how legal concepts like "substantial similarity" and "fair use" apply within the generative AI supply chain (Lee et al., 2023). See Appendix A.1

While there has been significant machine learning research on copyright detection, much of it focuses on identifying *if any* copyrighted works are detected in a (typically-vast) training corpus, rather than offering tools for understanding *how* or *to what extent* copyright might be being violated. More specifically, the problem is typically operationalized as recognizing whether a set of artwork(s) are present in, or substantially similar to, any data in the training corpus. We term this **general similarity detection (GSD)**, which includes problems like detecting exact copies of copyrighted work in a white-box corpus, or inferring their existence using the logits of pre-trained models.

This is a difficult and resource-intensive problem, ideally requiring white-box access to the training data and/or entire training procedure, making it largely inaccessible to anyone other than machine learning experts. Further, we note that this operationalization is not representative of the way copyright infringement cases are typically dealt with in court. These issues mean artists typically lack a technical basis for their claims, creating a legal bias towards the already-powerful tech companies releasing generative AI models.

In line with the legal concept of "differentiation" of copyrighted material, we frame the different and more tractable problem of **contextual similarity detection (CSD)**, which mirrors the real world setting where we know the reference class we're differentiating from – i.e. the similarity is contextualized by the choice of the reference set. In line with how the work would be used in a

legal setting, we use the legal jargon of **plaintiff set** for the query set, and **defendant set** for the potentially-infringing (e.g. AI-generated) reference set. We also propose a method for performing CSD where we fine-tune a pre-trained network and use averaged validation logits for the plaintiff class as the similarity score. In addition, we use saliency mapping, feature exploration, and other visualizations to ensure the model has captured differentiation-relevant features. See 1.1 for details. It is important to note the similarity score is not intended and should not be used as replacement for expert legal judgment. On the contrary, its purpose is to provide a narrow, specific, quantitative support to arguments that rest on substantial similarity, which fit into the broader context of a case with qualitative and other arguments.

Framing the quantification of substantial similarity as a classification problem among expert-identified sets has many advantages: (1) it makes use of expert knowledge and human judgement in assessing which classes are relevantly similar; (2) it thus allows the effective capacity of the model to be focused on features of the plaintiff class that are relevant for distinguishing it from similar artwork, rather than spread across the many possible categories if we were to consider the entire training corpus; (3) we can therefore use relatively low-resource models, accessible to ML non-experts.

The small size and customizability of our model are key features that make it practical and accessible for potential plaintiffs, including small studios and artists. Some intended use cases include: 1) public-interest litigators seeking to quantify infringement across cases of one or several artist clients; 2) individual artists seeking to defend their corpus of copyrighted material, 3) companies using AI-generated art seeking to assess the risk of infringement cases. Our research engages with the assumptions, theories and challenges underpinning ongoing legal battles, and contributes to the development of clearer guidelines and support tools for determining copyright infringement in AI-generated content.

### 1.1 FORMAL DEFINITIONS AND PROPOSED METHOD

Formally, **general similarity detection (GSD)** treats the problem of copyright infringement detection as pairwise similarity score estimation: letting $Q$ be the query set indexed by $i$, $X$ be the training corpus indexed by $j$, $\mathbb{1}$ be a vector-valued indicator function, and $\alpha$ be a threshold over which the images are considered similar, GSD seeks:

$$\mathbb{1}_Q = \begin{cases} 1 & \text{if } f(q_i, x_j) > \alpha \\ 0 & \text{otherwise} \end{cases} \tag{1}$$

with $f$ typically either an exact matching function such as a binary hash, or more often in recent years parameterized by a deep neural network pre-trained to perform image classification on a large corpus. $Q$ typically consists of a single image, and so GSD gives a binary 1 or 0 (yes or no) answer of whether or not the query image $q$ appears in the corpus $X$.

In contrast, in **contextualized similarity detection (CSD)**, expert human judgement is used to select a defendant subset $D \in X$ which contains one or more relevantly similar classes, and we seek a score for the plaintiff (query) set $\mathbb{S}_Q$ which quantifies its similarity to the defendant (reference) set:[1]

$$\mathbb{S}_Q = f(Q, D) \tag{2}$$

We anticipate and recommend that in CSD $|Q| = |D| << |X|$ (the size of the query and reference sets are approximately equal, and much smaller than the full corpus); this enables us to use very small models that can easily and cheaply be trained by non-experts.

The definition of CSD is very general; if the scoring function is performed (explicitly quantified or not) by an unaided human judge we simply have a mathematical description of a standard copyright case. However, it is common that judges' decisions are supported by relevant evidence and expert testimony, for example from artists familiar with similar works, and it is this type of evidence which our approach to CSD provides.

---

[1] While $f(Q, D)$ may be implemented by a neural network or other machine learning framework which treats images separately, and thus be very similar in practice to GSD, we use non-indexed notation here to emphasize that this is not necessary; plaintiff vs. defendant sets may be compared holistically.

We propose a methodology for CSD, which we call CSD-basic in our experiments (see Sec. 2). With this method we try to align as closely as possible with legal practice, while also making the most of pretrained representation learning methods in order to achieve good performance with relatively low resources. **Step 1:** Collect query (plaintiff) set of artworks. **Step 2:** Collect similarly-sized set or sets of reference (defendant) artworks, e.g. AI-generated and/or providing relevant elements of differentiation from the plaintiff artworks (e.g. Jerry is a cartoon mouse who resembles Mickey but is a different colour and shape). **Step 3:** Replace the last layer of a pretrained image classifier with classification among the plaintiff + defendant classes; separate the plaintiff + defendant dataset into train/validation sets. **Step 4:** With lower layers frozen, train the model – analogous to a human expert studying differences/similarities, we are training the last-layer representation to capture those similarities that are contextually relevant for both identifying and differentiating the classes. **Step 5:** Evaluate the model on the validation set, assessing model performance through accuracy, feature maps, misclassifications, visualizations, etc., and average the logit values for the plaintiff class across validation set. This score gives an overall idea of how much the model (specifically trained to distinguish defendent and plaintiff classes) finds the defendant set similar to the plaintiff.

## 1.2 RELATED WORK

Through the the mass growth of avenues for image sharing (search engines, social media, etc.), there has also been steady growth in the development of **technological solutions to detect and manage copyright breaches**. Google's Content ID automatically identifies and manages copyrighted audio and visual files on Youtube (Google, 2023). Kim et al. (2021) proposes a photo copyright identification framework that identifies copyright infringements of photos that have been manipulated through techniques like cropping, collages, or color changes. Scheffler et al. (2022) introduces a quantitative framework for assessing substantial similarity in copyright law, which uses easily-computed metrics such as description length, to quantitatively evaluate how derivative works may or may not be similar to their originals. Compared to these works, our approach is more technically novel and aligned with legal practice; the similarity score is computed based on a network trained to discriminate relevantly-similar works.

In using a trained neural network to estimte similarity, our work is similar to **CLIP**, a multimodality model trained on diverse internet-collected image-text pairs which assesses text-to-image similarity by comparing embeddings from both modalities (Radford et al., 2021). While using CLIP or other multimodality information on large automatically aligned corpora could be an interesting approach for future work, we focus on expert-identified relevantly similar classes in order to more closely duplicate the legal setting, as well as to make our approach amenable to small-data settings.

In **qualitative research**, there's been a growing discussion in how to attribute agency and authorship, and thereby infringement, in intellectual property law. One challenge is maintaining a consistent definition of these concepts across different mediums. Bellos & Montagu (2024), in their book *Who Owns this Sentence?*, provide a cultural, legal, and global history of the idea of copyright, explaining the concept of fair use and the difficulties in defining and applying it consistently across various contexts like music, art, and AI-generated content. Beyond authorship, there are difficulties in defining and applying copyright consistently throughout the AI supply chain specifically, as demonstrated by Lee et al. (2023). In this context, our work targets the "generation" phase of this supply chain, as opposed to building a tool that identifies infringing objects within the training data or analysing the prompt as the infringing object.

Finally, the **recent lawsuits** filed by artists Sarah Andersen, Kelly McKernan, and Karla Ortiz and several other artists against Stability AI, Midjourney, and other companies using Stability AI's Stable Diffusion models will likely set important precedents for how intellectual property law is applied to AI systems, potentially adhering to or differing from substantial similarity as a primary criterion. The plaintiffs in those cases contended that the defendants have unlawfully used their copyrighted works to train models without authorization. The United States District Court, in its tentative rulings from May 2024, decided not to dismiss the direct and induced copyright infringement claims, suggesting that courts may be open to offering greater protections to copyright holders whose works are used to train AI models without prior consent (United States District Court for the Northern District of California, 2024) – for instance, our method could be used to award damages proportional to substantial similarity.

## 2 EXPERIMENTS

We first present a **basic model** that assesses the similarity of images of cartoon mice generated by a Stable-Diffusion-xl model using prompts from Claude to stills of the 1928 Steamboat Willie version of Mickey Mouse. We made this choice because of the substantial case history surrounding Mickey (Disney has been an important figure in the development of copyright legislation), such that we have qualitative legal descriptions against which we can assess our quantitative findings. We train this model using four defendant classes, described below in Sec. 2.1 and visualized in Fig. 1b.

We perform a variety of hyperparameter explorations. Because a key tenet of our approach is the selection of relevantly similar defendant classes, we then explore whether and to what extent additional classes impact the model. Finally, we perform an analysis of the synthetic data and output the similarity scores using the optimal model hyperparameters and class specifications determined in the earlier experiments.

### 2.1 PLAINTIFF AND DEFENDANT DATASETS

The classes comprise of the following characters: characters from Warner Brothers' Foxy (a character designed by former Disney animators Hugh Harman and Rudolph Ising (IMDB, n.d.)), Van Beuren Studios' Milton the Mouse, and Hanna-Barbera's Tom and Jerry (see A.3 for images). Both Foxy and Milton serve as "ground truth" likenesses to Mickey Mouse, as Disney pursued legal action against Van Beuren Studios for Milton the Mouse's infringement on Mickey Mouse's likeness ((U.S. Court of Appeals for the Ninth Circuit, 1932)) (see Figure 1a for an annotated comparison of Milton and Mickey used in the original case Walt Disney Productions, Ltd. v. Pathe Exchange, Inc. and the Van Beuren Corporation (1932)). Disney won the case against Milton, while Warner Brothers discontinued Foxy's appearances after three shorts (Beck & Friedwald, 1989).

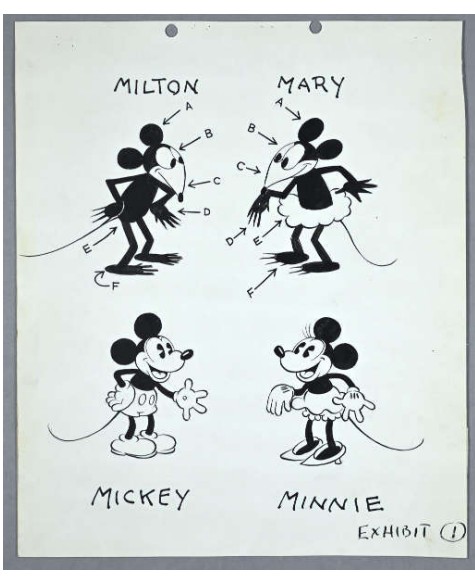

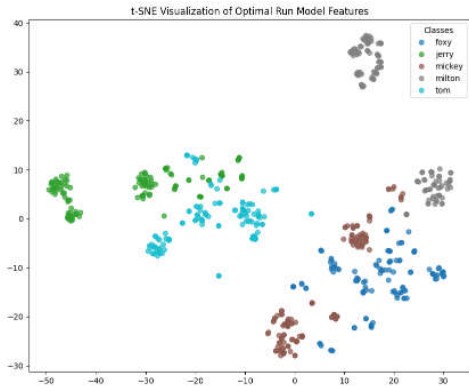

(a) Annotated comparison of Milton, and Mickey, Rita, and Minnie Mouse, from Walt Disney Productions, Ltd. v. Pathe Exhange, Inc. and the Van Beuren Corporation, illustrating the type of relevantly similar imagery used in legal cases.

(b) t-SNE visualization of trained model features across the plaintiff Mickey and defendant classes we selected: Foxy, Jerry, Milton, and Tom. The distinct separation of clusters indicates the model effectively differentiates between most classes, with each forming clear, dense groupings. However, the overlap observed between Foxy and Mickey (and to a lesser extent, Milton) highlights the substanital similarity between these three characters.

Figure 1: The two distinct clusters for Milton in the t-SNE (**right**) correspond to the original version of Milton and his later redesign to resemble Mickey, a change that led to Disney's copyright infringement lawsuit (**left**).

Tom and Jerry are included not only to evaluate model performance with color images, but also to provide an example of a cartoons that are not substantially similar to Mickey. Tom adds even more variability, expanding the context of "anthropomorphic cartoon animal" for the model. By including

these classes, we are essentially asking: "How similar to Mickey is the defendant set given the era of cartoon animals it belongs to?"

The dataset is small due to the limited number of images available for Foxy and Milton, whose careers were brief. The balanced dataset consists of 235 training images and 117 validation images for each class. Images of Foxy were obtained from the three Merrie Melodies shorts, while images of Milton were sourced from Aesop's Fables. The FFmpeg function was used to extract JPEGs from the public domain animated shorts. As is typical in small-data regimes, we apply data transformations (see A.2) to improve performance.

demonstraes

## 2.2 Hyperparameter tuning

We performed a series of experiments to investigate the impact of different hyperparameter settings (including weight decay, batch size, learning rate, et alia), which can often have a large impact on performace, particularly in low-data regimes such as those typically for copyright cases. We find that overall, performance is remarkably stable across hyperparameters, likely due to the combination of pre-trained network + context-relevant fine-tuning. This provides evidence that our proposed approach to CSD provides a strong baseline for other works, and that the hyperparameters we identify for the final run (Sec. A.7) will generalize to other contexts, which we validate in Sec. 3. See Appendix A.4 for full hyperparameter experiments.

## 2.3 Defendant class selection scaling experiments

### 2.3.1 Scaling: How does of content of classes affect performance?

In this experiment, we added additional classes of hand-drawn figures (see Figure 2) from the Google Quick, Draw! dataset to see if having varying drawn content in additional defendant classes would substantially change the model's performance. Each class in the Quick Draw dataset includes many different participants' drawings of the same prompt, so, in the "dog" class, the model was trained on many different variations of what an illustrated dog could look like.

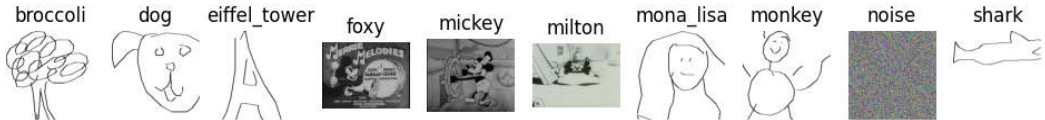

Figure 2: Classes used for Quick Draw experiment; content varies widely compared to the initial cartoon-mouse focused selected classes.

Figure 3 shows that any classification errors for Mickey, Foxy or Milton were with each other − the model distinguished these figures from the hand-drawn classes with ease, even though it did confuse the hand-drawn classes with each other.

Additionally, we experimented with distractor classes consisting of different noise distributions; see AA.10. Across these experiments we find evidence to support previous findings that low levels of noise improve generalization but high levels force the model to memorize. Because the performance (Figure 12) does not improve under any of the extra-content conditions, and there is no confusion across classes, we conclude that training the model with exclusively context-relevant classes is sufficient.

### 2.3.2 Scaling: How does the number of classes affect performance?

We investigate how performance varies as we scale the number of relevantly similar classes in the defendant set. The combination of three distractor classes—Milton, Foxy, and Jerry—yielded the highest validation accuracy, achieving a score of 0.9487 at Epoch 8 (see Figure 4). Similar validation scores were observed between two and four distractor classes. However, a significant decline in accuracy was noted when the number of distractor classes exceeded sixteen, particularly for 32,

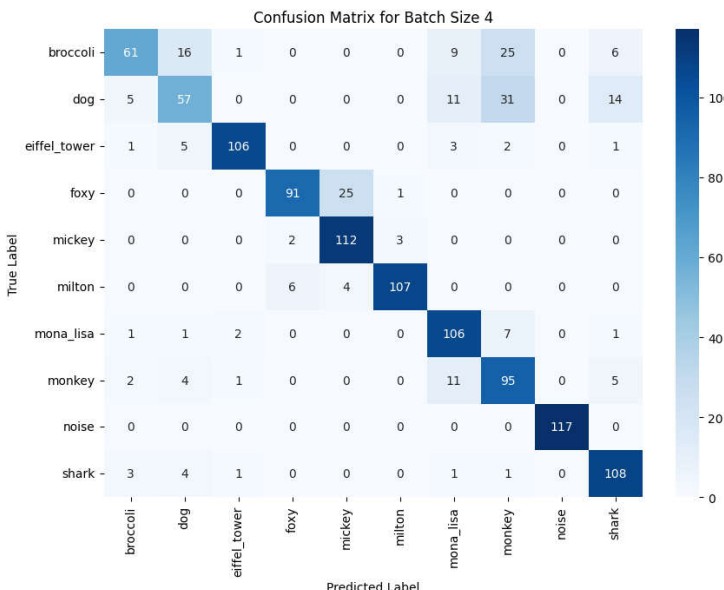

Figure 3: Confusion matrix for Quick Draw experiment, showing no confusion/overlap between Quick! Draw! classes and the originally selected defendant classes, suggesting that our legally-guided expert selection was successful in obtaining relevantly similar imagery and substantial content differentiation is unnecessary.

64, and 128 distractor classes. For optimal performance in novel applications of the model, it is recommended to use two to eight distractor classes, with three being the most effective.

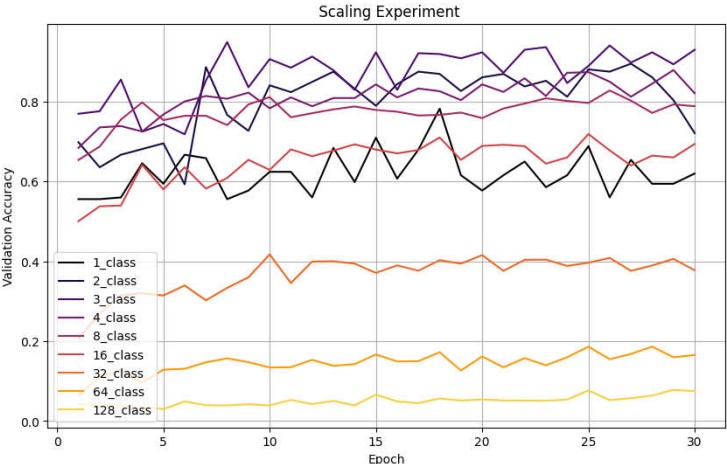

Figure 4: Validation accuracy across different numbers of classes across epochs, with between 2 and 8 classes performing the best.

## 2.4 FINAL RUN

Based on the results of our experiments, we ran the model with a batch size of 4, a weight decay of 0.001, a learning rate set at 0.001, and applied transformations (see A.2 for transformations) to the dataset of 5 classes (a number of classes that falls within the optimal range of 2 to 8 classes, per 2.3.2): Mickey, Milton, Foxy, Jerry, and Tom. A linear decay learning rate scheduler and an Adam optimizer with AmsGrad are employed with early stopping implemented with a patience of 10. The

model ran on a A100 GPU and converged at epoch 24 with a validation learning loss of 0.3206 and a validation accuracy of .9060 (see Table 1; see detailed performance plots in A.7).

| Metric | Train | Validation |
|---|---|---|
| Loss | 0.4704 | 0.3206 |
| Accuracy | 0.8272 | 0.8991 |

Table 1: Performance metrics of the final run for the training and validation sets, demonstrating that the model has captured enough relevant information to achieve good performance.

For Output 1 (see Figures 5a and 5b), the image is quantified **81.2% similar**. This shows that the model was able to recognize the "ground truth" in the context of copyright infringement of Steamboat Willie-era Mickey Mouse style (which was copied in Milton and Foxy as well). Like the feature visualizations in the basic model, the ears, body, tail and nose shape of this image are all identified throughout as being significant indicators of stylistic similarity.

For Output 2 (see Figures 5c and 5d), the image is quantified **2.5% similar**. Figure 5c shows that the model is confounded by the content within the circle, but identifies the torso of the mouse as bearing some resemblance to Mickey (perhaps due to the fact that this is the only area in the image with the stark black and white contrast we see on Mickey's body). The ears, facial expression and feet aren't included in the matching, which is where the image differs significantly in style to Mickey.

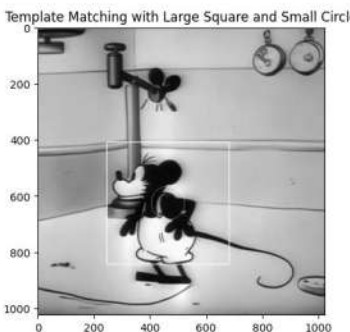

(a) Template matching for output 1, which bears a 0.812 probability that it reproduces Mickey Mouse

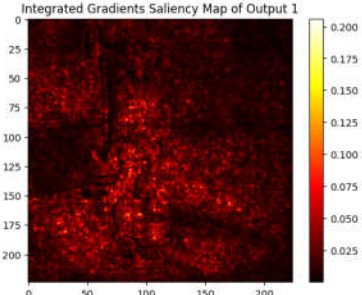

(b) Saliency map for output 1, which bears a 0.812 probability that it reproduces Mickey Mouse

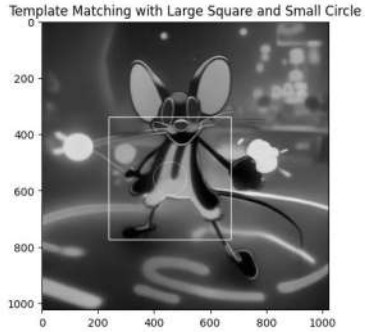

(c) Template matching for output 2, which bears a 0.025 probability that it reproduces Mickey Mouse

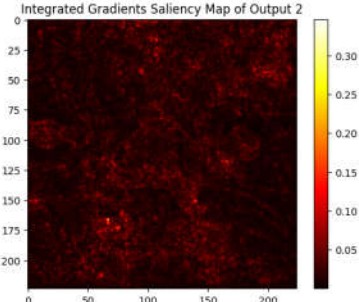

(d) Saliency map for output 2, which bears a 0.025 probability that it reproduces Mickey Mouse

Figure 5: Analyses for final run outputs 1 and 2.

## 3 GENERALIZATION OF EXPERIMENTAL METHODOLOGY: PRYMACHENKO

To demonstrate our approach's generalizability, we apply it to the Ukrainian artist Maria Prymachenko's body of work as the plaintiff set, and AI-generated images . Being another example of hand-drawn images with similar composition, we wanted to explore how the model generalizes to a markedly different style.

Maria Prymachenko's style of folk art features animals, people, and other creatures that are styled with distinctive, repetitive patterns, making it a source of inspiration for surface designs. Her work, which is housed by the National Museum of Ukrainian Folk Art, was plagiarized by Marimekko, a Finnish clothing and home furnishing company, and Finnair, Finland's largest airline, which used the print on their aircrafts in a collaboration with Marimekko (Center for Art Law, 2013), again giving us qualitative legal arguments in which to contextualize our quantified scores.

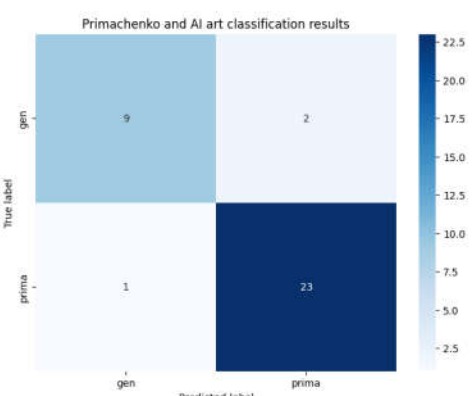

(a) Classification results for Prymachenko art and AI generated art

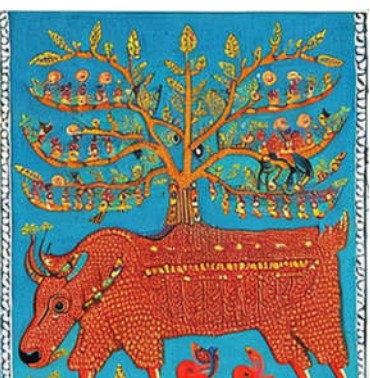

(b) False positive 1 for Prymachenko art (is AI)

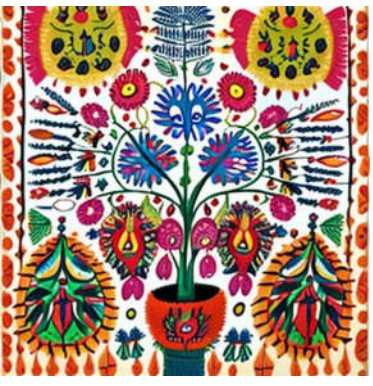

(c) False positive 2 for Prymachenko art (is AI)

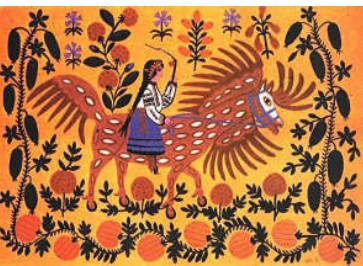

(d) False positive for AI art (is Prymachenko)

Figure 6: Confusion matrix and exploration of false positives

We generated AI dupes of her work by fine-tuning a Stable Diffusion Realistic Vision V1.3 on 120 images of her artwork, using prompts generated by Claude 3 Opus to dictate the content of each image. The images of her work were scraped from the collection publicly available on WikiArt (WikiArt, 2024). This provides an example of scenario where a plaintiff may not have context-relevant images to compare their own work to, but have access to a model that they believe has trained on their work enough to reproduce copyrightable elements. Table 2 and Figure 6a summarize the performance results on the validation set, including a 91% accuracy score.

The AI-generated false positives (Figures 6b and 6c,which were classified as Prymachenko's originals) are telling – the motif of an animal in the foreground with symmetrical, repeated patterns

Table 2: Performance results on validation set for Prymachenko vs AI-art classification

| Class | Precision | Recall | F1-Score |
|---|---|---|---|
| gen | 0.90 | 0.82 | 0.86 |
| prima | 0.92 | 0.96 | 0.94 |
| **Accuracy** | | 0.91 | |

surrounding it are typical of the kind of visual storytelling in her work. She also has series of flowers in vases, which was also copied by the generative AI model and similar enough to be confused as her own work.

The Prymachenko-original false positives (Figure 6d, which was classified as AI) reflects patterns in the training data – a lot of the AI-generated artworks feature a horse and rider in the center of the frame.

The AI set in its entirety bears a 0.687 similarity score, i.e. is **68.7% similar** to Prymachenko's work (this figure being the average of the softmax probabilities for the Prymachenko class). In conjunction with the styles depicted in the false positives (Figures 6b and 6c), our findings suggest that the AI set is infringing on the copyright of Prymachenko's style.

## 4 CONCLUSION

Copyright infringement is of increasing concern in an era of AI-generated content, particularly given the power imbalance between artists and the large tech corporations creating generative AI systems. Our work sits at the intersection of technical and legal scholarship, providing conceptual clarity, benchmarks, and practical tools for both machine learning researchers in this nascent field as well as for real-world legal cases.

### 4.1 CONTRIBUTIONS OF THIS WORK

1. Surveying of relevant legal context and presentation of relevant legal concepts such as differentiation and substantial similarity

2. Formal description of the operationalization of this problem in machine learning - general similarity detection (GSD) and introduce contextualized similarity detection (CSD); more in line with real legal contexts by assessing similarity of a set of plaintiff works (e.g. by an artist) to a defendant set (e.g. AI-generated and/or contextually relevant).

3. Proposal, experimental validation, and qualitative legal correspondence analysis of a method for CSD (CSD-basic) involving pretrained image models and fine-tuning on a variety of expert-selected relevantly similar classes.

4. Experimental validation of the generalization of CSD-basic to a novel setting.

### 4.2 FUTURE WORK

We will be running a legal clinic, bringing together public interest litigators, machine learning researchers, and artists, to apply and expand on our approach. As part of this initiative, we plan to test the approach with a variety of other datasets. In particular, while this work focuses on images, we would like to consider and test for other media, especially sound/music.

In addition to the copyright litigation applications we have primarily targeted, our approach could also be used in remuneration schemes, like that proposed by ProRata.ai (Mok, 2024). Many of Prymachenko and other artists' original works were tragically destroyed during the Russian invasion of Ukraine. For art that is digitally preserved, aside from living artist compensation, methods like ours could provide ways to chart stylistic influence and thereby raise funds for restoration or legacy promotion projects that are particularly important when originals are lost. Automated methods to select relevantly similar defendant classes could be employed to scaled up this approach across many plaintiff classes, to support class action lawsuits.

We hope our work inspires more ML researchers to pursue interdisciplinary techno-legal works, such as clearer, technically relevant guidelines on determining copyright infringement in AI-generated material, more robust and consistent standards for evaluating potential infringement, and methods to empower and remunerate artists.

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
