## A   APPENDIX

### A.1   LEGAL CONTEXT

The concept of "substantial similarity" is central to determining copyright infringement, but there is no "bright-line" rule (i.e. clear and producing consistent results) for establishing it. Courts often consider factors such as the "total concept and feel" of the works in question and the level of creativity involved in the copyrighted work (first defined in U.S. Court of Appeals for the Ninth Circuit (1977)), in conjunction with expert testimony and analysis.

To establish copyright infringement, a plaintiff must first demonstrate that the alleged infringer actually used the copyrighted work in their purportedly infringing activities. Sometimes plaintiffs have direct evidence that the alleged infringer used their copyrighted work in the defendant's purportedly infringing activities. For instance, a defendant may admit that the copyrighted work was their inspiration in creating their own work. Or perhaps the plaintiff can point to eyewitnesses of the alleged copying. But often, perhaps typically, direct evidence is lacking. When it is lacking, courts may consider a combination of (1) evidence of the defendant's access to the copyrighted work; and (2) similarities between the defendant's work and the original copyrighted work that suggests copying, in determining whether the alleged infringer actually copied from the copyrighted work.

Two works are substantially similar when "the ordinary observer, unless [they] set out to detect the disparities, would be disposed to overlook them, and regard their aesthetic appeal as the same" (U.S. Court of Appeals for the Second Circuit, 1960). A common test is a "holistic, subjective comparison of the works to determine whether they are substantially similar in total concept and feel" (U.S. Court of Appeals for the Ninth Circuit, 2018).

Historically, some courts have dispensed with the requirement for evidence of access if the works are so "strikingly similar" that it is more likely than not that copying occurred (U.S. Court of Appeals for the Second Circuit, 1946). Interestingly, the Ninth Circuit has recently retired the related inverse ratio rule - the concept that as evidence of access increases, the evidentiary threshold for identified similarity to prove copying decreases, and vice versa. In light of this change, it is possible that other circuits may follow suit in the future to maintain consistency in jurisprudence relating to copyright.

When assessing substantial similarity, courts often employ the "extrinsic-intrinsic test," which was first articulated in *Sid & Marty Krofft Television Productions, Inc. v. McDonald's Corp.* (U.S. Court of Appeals for the Ninth Circuit, 1977). The extrinsic component of the test involves an objective analysis of the similarities in ideas and expression between the works, while the intrinsic component is a more subjective assessment of overall similarities from the perspective of the "ordinary reasonable person" (or a similar description of a reasonable individual possessing no related expert knowledge) (See U.S. Court of Appeals for the Ninth Circuit (2004); U.S. Court of Appeals for the Ninth Circuit (1994); U.S. Court of Appeals for the Ninth Circuit (1986)). Although expert testimony may be considered in the extrinsic analysis, it is inappropriate for the intrinsic test due to its focus on the perspective of the ordinary person (U.S. Court of Appeals for the Ninth Circuit (1988); U.S. Court of Appeals for the Ninth Circuit (2016)). Further, the application of the substantial similarity test may vary depending on the subject matter and medium of the works in question.

By providing a quantitative assessment of substantial similarity, frameworks like ours can aid plaintiffs in defending their intellectual property rights by offering explicit similarity metrics, tailored to individual contexts. In the future, combining computational tools with interpretability methods may make it possible to identify where infringement occurs in the training and generation process, thereby identifying a new kind of "infringing object", which would advance our understanding of how to apply copyright protection in the context of generative AI.

That being said, these tools should not be viewed as a replacement for expert analysis and legal judgment. The concept of substantial similarity is inherently complex and context-dependent, and courts have emphasized the importance of considering the "total concept and overall feel" of the works in question, rather than relying on mechanical dissection or quantitative measures alone (U.S. Court of Appeals for the Ninth Circuit, 1977; 2018). Further, these tools are not sufficient to support the practice of law without a relevant education or bar membership.

## A.2 Transformations

| Transformation | Parameters |
|---|---|
| RandomResizedCrop | 224 (size) |
| RandomHorizontalFlip | None (default probability of 0.5) |
| RandomRotation | 10 (degrees) |
| ColorJitter | brightness=0.2, contrast=0.2, saturation=0.2, hue=0.1 |
| RandomAffine | degrees=0, translate=(0.1, 0.1) |
| ToTensor | None |
| RandomPerspective | distortion_scale=0.05, p=0.5 |
| RandomErasing | p=0.1, scale=(0.02, 0.33), ratio=(0.3, 3.3), value=0, inplace=False |
| Normalize | mean=[0.485, 0.456, 0.406], std=[0.229, 0.224, 0.225] |

## A.3 Classes

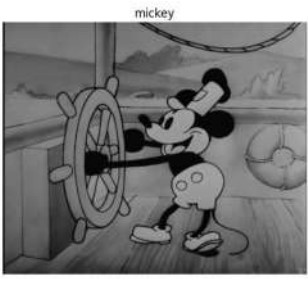
(a) Mickey

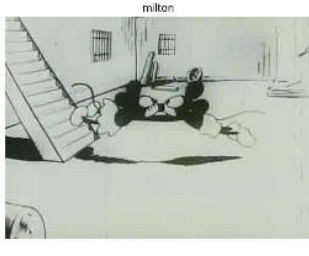
(b) Milton

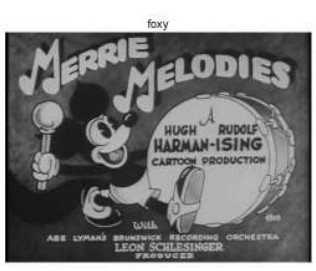
(c) Foxy

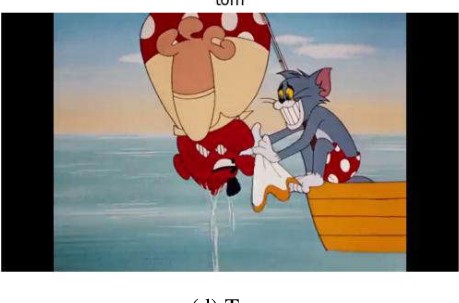
(d) Tom

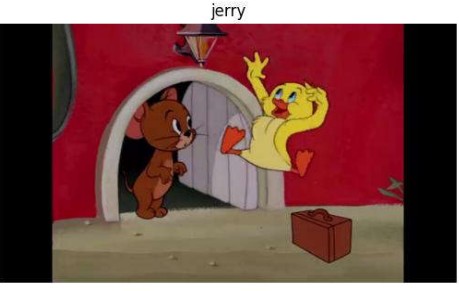
(e) Jerry

Figure 7: Collection of characters.

## A.4 Hyperparameter Tuning Experiments

### A.4.1 Batch Size and Weight Decay Experiment

We applied the following combinations of batch sizes and weight decays to identify the best combination for our model: batch size 4 and weight decay 0.01, batch size 6 and weight decay 0.01; batch size 4 and weight decay 0.001, batch size 6 and weight decay 0.001; batch size 4 and weight decay 0.0001, batch size 6 and weight decay 0.001.

We observed the best performance using a batch size of 4 with a weight decay of 0.001 (see Table 3, and see all combinations classification reports in A.4).

## A.5 Learning Rate Experiments

Based on the findings by Li et al. (2020), we used a Linear Decay Learning Rate Scheduler. The implementation of a Linear Decay Learning Rate was shown to be beneficial for a ResNet model constrained by a fixed resource budget, offering a simple, robust, and high-performing compared

Table 3: Classification report for batch size 4, weight decay 0.001

|  | precision | recall | f1-score | support |
|---|---|---|---|---|
| **Foxy** | 0.91 | 0.92 | 0.92 | 117 |
| **Jerry** | 0.81 | 0.96 | 0.88 | 117 |
| **Mickey** | 0.94 | 0.87 | 0.90 | 117 |
| **Milton** | 0.95 | 1.00 | 0.97 | 117 |
| **Tom** | 0.95 | 0.78 | 0.85 | 117 |
| **accuracy** | | | 0.91 | 585 |
| **macro avg** | 0.91 | 0.91 | 0.91 | 585 |
| **weighted avg** | 0.91 | 0.91 | 0.91 | 585 |

to other learning rate schedules (Li et al., 2020). This is crucial for our model, especially since we aim for real-world applications where limited computational resources are available to creative and legal stakeholders. This approach systematically reduces the learning rate in proportion to the total iteration budget, which is especially effective under resource-constrained settings. Compared to our previous runs with StepLR schedulers, the linear decay learning rate has proven to be more effective lowering our validation learning loss and increasing validation accuracy.

A Learning Rate Find was performed to locate an optimal learning rate for the model, with a learning rate between 0.001 and 0.01 emerging as ideal (Figure 8). The learning rate has been set at 0.001 for previous experiments and this plot confirms that value is optimal for our model.

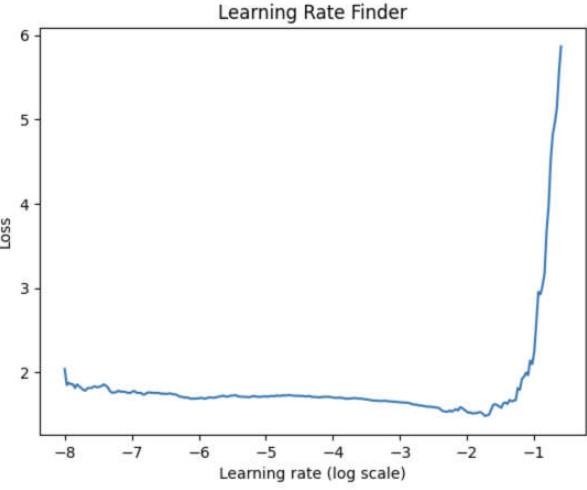

Figure 8: Optimal learning rate range plot.

A.6 CROSS-VALIDATION

We performed a 5-fold cross-validation to test our model's validation accuracy and validation learning loss. We found that a batch size of 6 on the fourth fold yielded the best performance, with an accuracy of 0.84, and overall yielded a strong and stable performance that almost matched runs using a batch size of 4 (see Figure 9). We include both batch sizes 4 and 6 in our analysis, which, given the stochastic nature of each batch size, will allow for more interpretation of accuracies across both batch sizes.

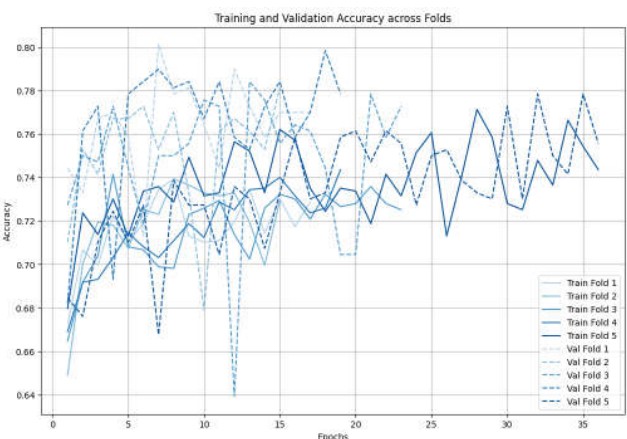

Figure 9: Cross-validation performance using a batch size of 4.

**Batch size 6 with weight decay 0.001:** We observed similar results to Batch Size 4, however, the precision and recall of the Mickey class are imbalanced compared to Batch Size 4 and we have a lowered overall accuracy of the validation set.

Table 4: Classification report for batch size 6 with Weight Decay 0.001

|              | precision | recall | f1-score | support |
|--------------|-----------|--------|----------|---------|
| **Foxy**     | 0.78      | 0.95   | 0.86     | 117     |
| **Jerry**    | 0.80      | 0.95   | 0.87     | 117     |
| **Mickey**   | 0.94      | 0.69   | 0.80     | 117     |
| **Milton**   | 0.95      | 1.00   | 0.97     | 117     |
| **Tom**      | 0.94      | 0.76   | 0.84     | 117     |
| **accuracy** |           |        | 0.87     | 585     |
| **macro avg**| 0.88      | 0.87   | 0.87     | 585     |
| **weighted avg** | 0.88  | 0.87   | 0.87     | 585     |

**Batch size 4 with weight decay 0.0001:** We observed that while the accuracy remained constant to Batch Size 4 with a Weight Decay of .001, a concern of overfitting remains due to the small size of the data set, therefore a more robust weight decay of .001 is preferred moving forward.

Table 5: Classification report for batch size 4 and weight decay of 0.0001

|              | precision | recall | f1-score | support |
|--------------|-----------|--------|----------|---------|
| **Foxy**     | 0.91      | 0.92   | 0.92     | 117     |
| **Jerry**    | 0.81      | 0.96   | 0.88     | 117     |
| **Mickey**   | 0.94      | 0.87   | 0.90     | 117     |
| **Milton**   | 0.95      | 1.00   | 0.97     | 117     |
| **Tom**      | 0.95      | 0.78   | 0.85     | 117     |
| **accuracy** |           |        | 0.91     | 702     |
| **macro avg**| 0.91      | 0.91   | 0.91     | 585     |
| **weighted avg** | 0.91  | 0.91   | 0.91     | 585     |

**Batch size 6 with weight decay 0.0001:** We observed worse overall accuracy, recall and precision for the Mickey class.

Table 6: Classification report for batch size 6, weight decay 0.0001

|  | precision | recall | f1-score | support |
|---|---|---|---|---|
| **Foxy** | 0.79 | 0.95 | 0.86 | 117 |
| **Jerry** | 0.80 | 0.95 | 0.87 | 117 |
| **Mickey** | 0.94 | 0.70 | 0.80 | 117 |
| **Milton** | 0.95 | 1.00 | 0.97 | 117 |
| **Tom** | 0.94 | 0.76 | 0.84 | 117 |
| **accuracy** |  |  | 0.87 | 702 |
| **macro avg** | 0.88 | 0.87 | 0.87 | 585 |
| **weighted avg** | 0.88 | 0.87 | 0.87 | 585 |

**Batch size 4 with weight decay 0.01:**   This setting provided no improvement for the Mickey class, overall accuracy, nor did it lower the validation learning loss.

Table 7: Classification report for batch size 4, weight decay 0.01

|  | precision | recall | f1-score | support |
|---|---|---|---|---|
| **Foxy** | 0.91 | 0.91 | 0.91 | 117 |
| **Jerry** | 0.81 | 0.96 | 0.88 | 117 |
| **Mickey** | 0.93 | 0.87 | 0.90 | 117 |
| **Milton** | 0.95 | 1.00 | 0.97 | 117 |
| **Tom** | 0.95 | 0.77 | 0.85 | 117 |
| **accuracy** |  |  | 0.90 | 585 |
| **macro avg** | 0.91 | 0.90 | 0.90 | 585 |
| **weighted avg** | 0.91 | 0.90 | 0.90 | 585 |

**Batch size 6 with weight decay 0.01:**   We observed a loss in the recall and precision balance seen in the previous setting for the Mickey class and a lowered overall accuracy.

Table 8: Classification report for batch size 6, weight decay 0.01

|  | precision | recall | f1-score | support |
|---|---|---|---|---|
| **Foxy** | 0.78 | 0.95 | 0.86 | 117 |
| **Jerry** | 0.78 | 0.93 | 0.85 | 117 |
| **Mickey** | 0.90 | 0.71 | 0.79 | 117 |
| **Milton** | 0.97 | 0.97 | 0.97 | 117 |
| **Tom** | 0.92 | 0.74 | 0.82 | 117 |
| **accuracy** |  |  | 0.86 | 585 |
| **macro avg** | 0.87 | 0.86 | 0.86 | 585 |
| **weighted avg** | 0.87 | 0.86 | 0.86 | 585 |

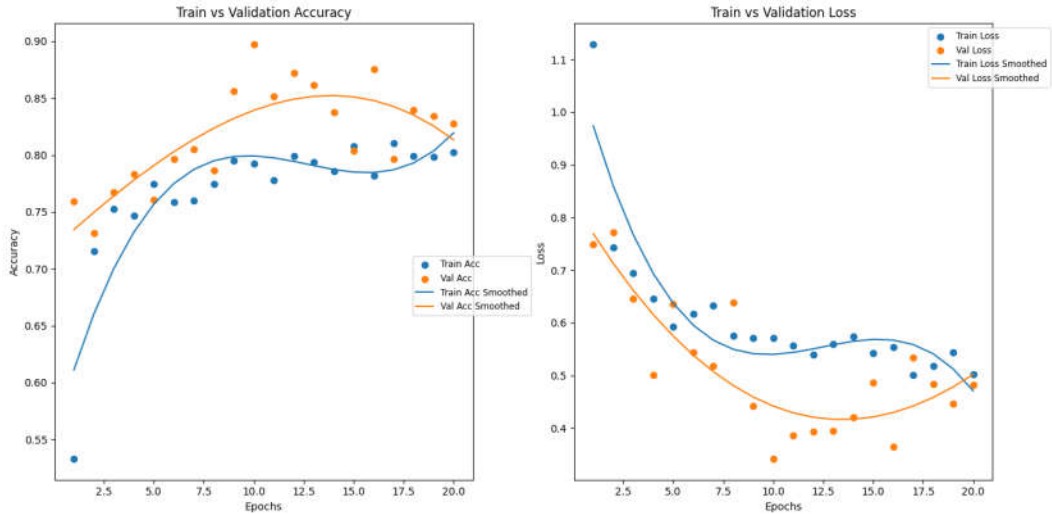

Figure 10: Final run of Mickey experiments performance, using a batch size of 4, a weight decay of 0.001, a learning rate set at 0.001, and applied transformations to the dataset of 5 classes.

## A.7    FINAL RUN PERFORMANCE PLOTS

## A.8    FEATURE MAPS

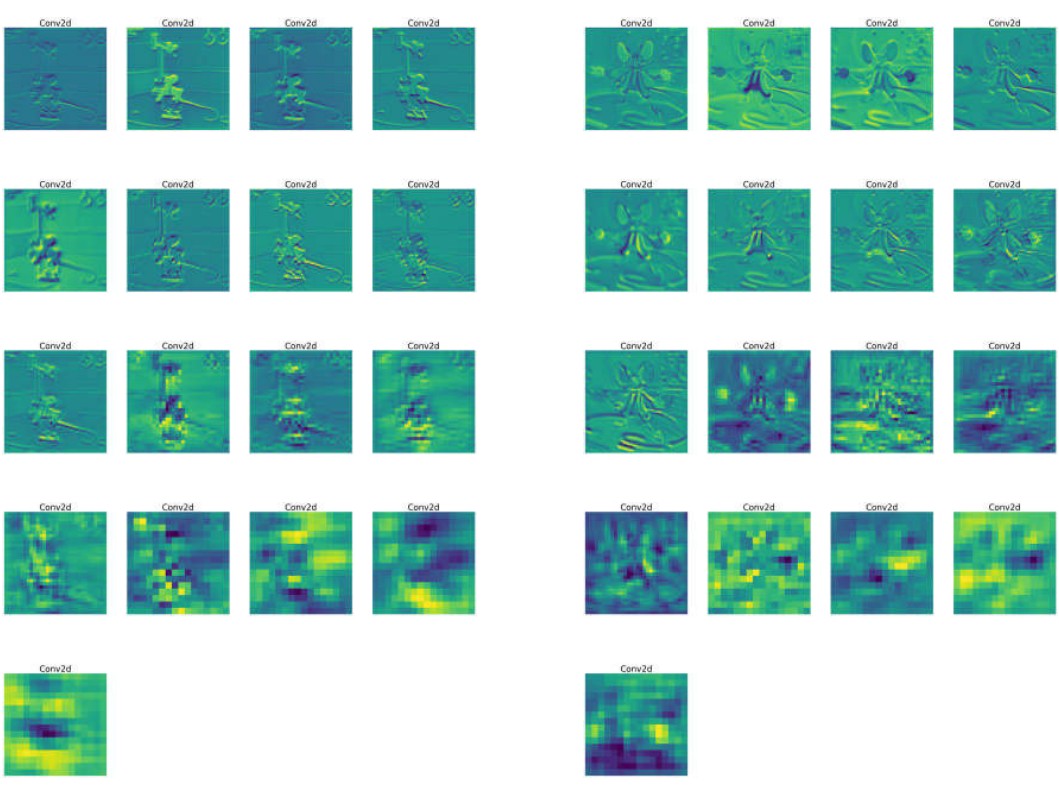

(a) Feature map for output 1.                    (b) Feature map for output 2.

Figure 11: Feature maps for final run of Mickey experiments, using a batch size of 4, a weight decay of 0.001, a learning rate set at 0.001, and applied transformations to the dataset of 5 classes..

A.9   ADDITIONAL QUICK! DRAW! EXPERIMENTS

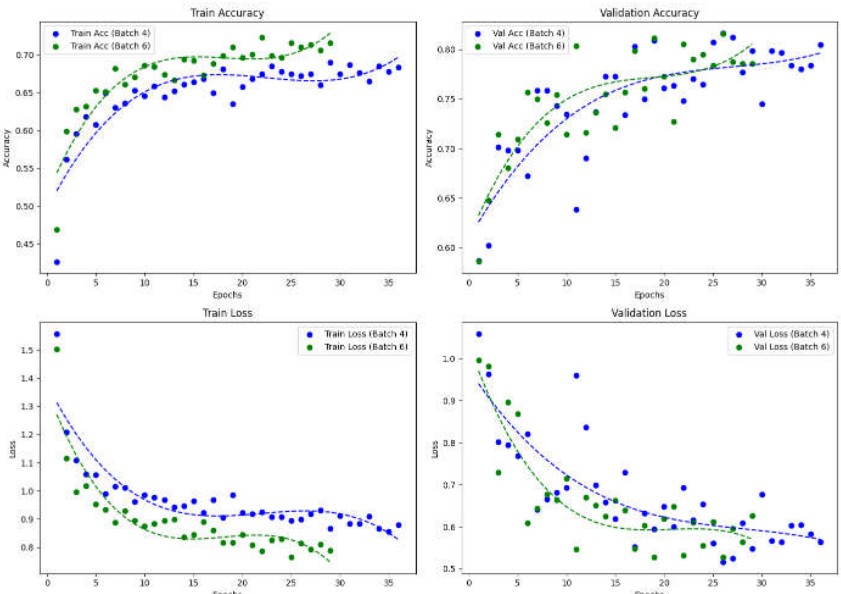

Figure 12: Performance plot for Quick Draw experiment.

A.10   NOISE PLOTS

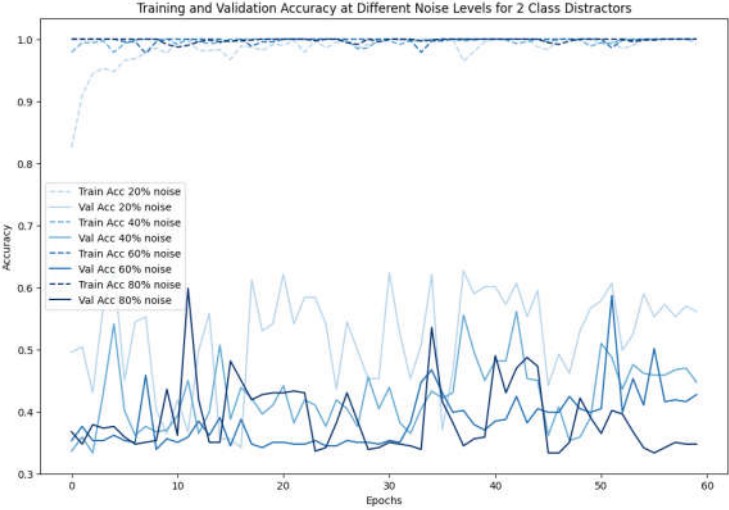

Figure 13: Scaling to two additional distractor classes.

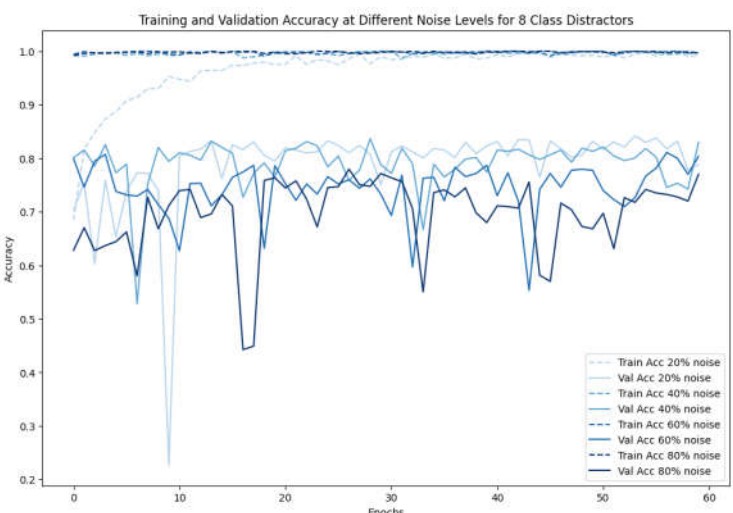

Figure 14: Scaling to eight additional distractor classes.

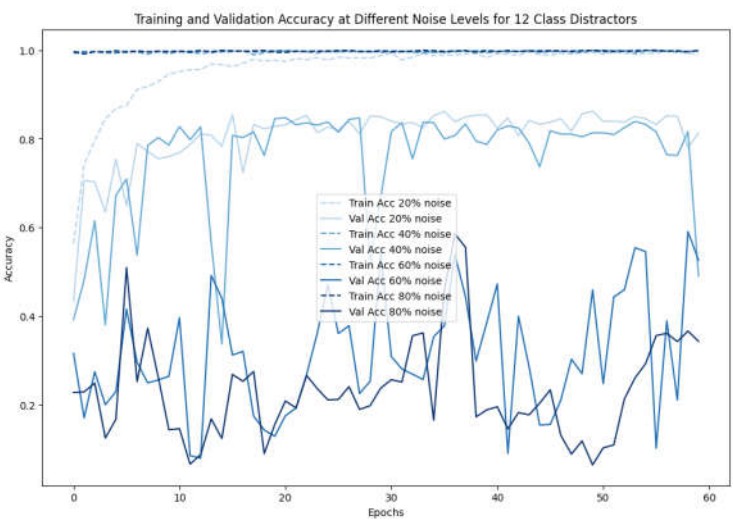

Figure 15: Scaling to 16 additional distractor classes.