# OpenReview forum: "Quantifying Likeness: A Simple Machine Learning Approach to Identifying Copyright Infringement in (AI-Generated) Artwork"
_ICLR.cc/2025/Conference — ICLR 2025 Conference Withdrawn Submission_

### Official Review · Reviewer_nBdG · 2024-10-20

**Soundness:** 2
**Presentation:** 1
**Contribution:** 1
**Rating:** 1
**Confidence:** 5

**Summary:**

This paper formulates the problem of identifying copyright infringements in AIGC artworks. Specifically, it exploits a classifier network to detect generated images with copyright contents.

**Strengths:**

This paper focuses on a meaningful topic: the copyright infringement of AIGC applications.

**Weaknesses:**

This paper suffers from several main defects:

1) Motivation: one claimed contribution of this paper is to formulate the copyright infringement identification as a machine learning problem. Unfortunately, this problem is not practical. For the plaintiffs, it is straight-forward to recognize similar AIGC contents to their own artworks. For the court, it is the judge and the expert who determine whether there are substantial similarities between AIGC artworks and the defendant's artworks, which is the core part of the court session. Hence, neither the plaintiffs nor the judge need an extra identifier.

2) Effectiveness: this method only shows its effectiveness in the very simple Mickey Mouse dataset, where the copyright figure is too simple to be identified by humans. Also, the method requires training a new classifier for identifying new copyright figures, meaning that it is not generalizable at all.

3) Novelty: the method only exploits a simple classifier without any novel designs.

4) Presentation: the paper seems to be incomplete, e.g. "demonstraes" in line 225.

In general, this paper adapts an off-the-shelf method to an unrealistic problem, without enough evidence of effectiveness. I think it is generally meaningless.

**Questions:**

Please address the above issues on the motivation, the effectiveness, and the novelty.

---

### Official Review · Reviewer_tJCY · 2024-10-21

**Soundness:** 3
**Presentation:** 2
**Contribution:** 2
**Rating:** 3
**Confidence:** 4

**Summary:**

The paper "Quantifying Likeness: A Simple Machine Learning Approach to Identifying Copyright Infringement in (AI-Generated) Artwork" presents a framework to quantify stylistic similarity between AI-generated and copyrighted artworks, aligning with legal precedents. Using a method called contextual similarity detection (CSD), the authors fine-tune neural networks to compare infringing (defendant) works against copyrighted (plaintiff) works, with a focus on widely recognized characters like Mickey Mouse. They validate the approach through experiments and argue its relevance for copyright litigation, providing a practical tool to support legal experts in assessing substantial similarity in AI-generated content. The method shows potential for broader applications across media, helping address copyright challenges in the era of generative AI.

**Strengths:**

1. **Originality:** The paper introduces a novel *contextual similarity detection (CSD)* method tailored to legal contexts, offering a creative adaptation of machine learning to copyright infringement detection, particularly in AI-generated content.

2. **Significance:** The method has practical implications for copyright litigation, providing a quantitative, legally aligned tool that can aid in assessing substantial similarity in AI-generated works, with potential for broader media applications.

**Weaknesses:**

**Weaknesses:**

1. **Lack of Comparison with Similar Work**: A key technical oversight is the omission of relevant literature, particularly Moayeri et al.'s work titled "Rethinking Artistic Copyright Infringements in the Era of Text-to-Image Generative Models" [1]. This paper, which introduces *ArtSavant* for detecting style copying in AI-generated art, addresses a similar problem of quantifying artistic style infringement. Both papers focus on style comparison in a legal framework, yet the authors of this paper do not discuss or compare their method with *ArtSavant*, missing an opportunity to clarify distinctions, improvements, or complementary aspects of their approach. Incorporating such comparisons would strengthen the novelty and positioning of their method.

2. **Limited Dataset for Validation**: The paper uses relatively small datasets, focusing on iconic characters like Mickey Mouse and Maria Prymachenko’s art. While these serve as high-profile examples, they may not generalize well to a broader range of artistic styles, especially in non-animated or modern contexts. For a robust evaluation, a wider variety of artistic styles from different time periods, genres, and media should be incorporated. Expanding the dataset and performing more diverse tests would improve confidence in the method’s scalability and real-world applicability.

3. **Lack of Interpretability in the Method**: The paper’s reliance on neural network logit scores, while effective for classification, lacks the necessary interpretability for legal use, where explainability is crucial. Methods like the *TagMatch* approach in Moayeri et al.'s work [1], which provides interpretable tag-based signatures, would make the results more understandable and actionable for legal professionals. Adopting a more interpretable framework, such as combining neural outputs with human-understandable tags or visual explanations, would greatly enhance the usability of the model in court settings, where the reasoning behind decisions must be transparent and easily explainable to non-experts.

[1] Moayeri, M., Basu, S., Balasubramanian, S., Kattakinda, P., Chengini, A., Brauneis, R., & Feizi, S. (2024). Rethinking Artistic Copyright Infringements in the Era of Text-to-Image Generative Models. arXiv preprint arXiv:2404.08030.

**Questions:**

I encourage the authors to address the points raised in the weaknesses section and to conduct additional experiments where further investigation is required.

---

### Official Review · Reviewer_iygy · 2024-10-27

**Soundness:** 1
**Presentation:** 1
**Contribution:** 2
**Rating:** 3
**Confidence:** 5

**Summary:**

The work proposes a ML based approach to quantify copyright infringement in AI-generated artwork by assessing stylistic similarity. It argues that existing copyright detection methods are not practical for real-world legal scenarios and propose a more accessible and customizable model for artists to evaluate the likelihood of infringement. The study uses case studies involving Mickey Mouse and Maria Prymachenko's work to illustrate the model’s potential and includes experiments to assess the model’s performance, robustness, and hyperparameters.

**Strengths:**

The problem of creating a tool for artists for helping determine potential copyright violations is interesting and worth exploring.

**Weaknesses:**

**Non-standard pdf**: The file seems to have been run through some pdf flattening software which has disabled text selection, highlighting, and hyperlinking, which makes it difficult to follow.

**Insufficient engagement with ML literature**: There are only 24 citations in the bibliography, with more than half pertaining to legal literature. There are only around 5 references to ML papers. In my opinion, the work does not sufficiently engage with existing ML literature to justify acceptance in a ML centered conference.

**No rigorous benchmarking**: The paper seems to be driven by case studies with no rigorous benchmarking. The results of the experiments described in the paper are of unclear significance.

**Limited novelty**: I could not make out any significant, concrete contributions in the paper from an ML perspective, even granting that the work is centred around operationalizing copyright law using ML.

In general, the work seems unfinished and not suitable for presenting at a conference. I urge the authors to submit their work to a workshop to gain more feedback and improve the manuscript.

**Questions:**

NA

---

### Official Review · Reviewer_4Ao4 · 2024-11-03

**Soundness:** 2
**Presentation:** 2
**Contribution:** 3
**Rating:** 3
**Confidence:** 5

**Summary:**

The authors present a framework for quantifying similarity of two sets of artwork (e.g. real and AI generated) with implications and grounding in legal copyright scholarship. They propose to tune a classifier to distinguish "defendant" and "plaintiff" (along with some "distractor") classes of art, and use the average softmax probability scores as a metric for similarity. They show some level of stability to hyperparameters (while noting that performance can change depending on the number of distractor classes) and suggest saliency maps can add qualitative insight atop their quantitative score. Two case studies are demonstrated, inspired by real legal cases.

**Strengths:**

The problem is significant -- its sociotechnical nature makes it uniquely challenging and impactful to a wide audience.

The legal grounding is excellent. Many cases are discussed, as well as existing tenets of copyright law. The nomenclature of 'defendant' and 'plaintiff' sets is appealing, and may make the work more accessible to legal stakeholders.

The approach diverges from typical (existing) image similarity methods -- which the authors frame as 'generalized similarity detection' (vs. their 'contextualized similarity detection') -- is important.

I think a simpler method, like this classifier based approach, is more practically accessible than a more technically involved one -- to me, the simplicity is a strength.

**Weaknesses:**

While the idea and motivation are great, the execution is underwhelming:
- The experiments are very limited and somewhat inconclusive. Only two cases are studied, and I am not sure if ultimately the method provides a clear judgement on if copyright was infringed or if substantial similarity is met.
- I do not understand how the "AI set in its entirety bears a 0.687 similarity to Prymachenko's work" (L448), when the precision and recall for the AI set is high (table 2). If the AI images are consistently being classified to the AI class, then wouldn't the softmax probability for the other class (Prymachenko) be consistently low (at least <0.5)?
- The experiments show a surprisingly large drop in accuracy when incorporating more distractor classes. Are all distractor classes coming from the quick draw dataset? This is a questionable choice in my opinion -- comparing to other art or cartoon characters would be more apt. Also, it seems like the model may not be training correctly -- validation accuracy barely increases for the 128 class case; maybe a different learning rate is needed for that task.


Comparing to quick draw classes does not feel appropriate -- I'd image something as simple as taking the average pixel value could suffice in distinguishing the cartoon characters from the quick draw images (since the cartoons have shading while quick draw does not). The low classifier accuracy for high class counts could simply be a result of quick draw classes being very similar, which is not the important comparison (i.e. between defendant vs plaintiff classes).

The clarity/presentation could be much better -- aside from typos (see below), many details are omitted, like the size of the classes that the classifier is trained on, and crucially, how one should interpret the outputs of your system: what similarity score would suggest copyright infringement, and why?

Use of saliency maps is questionable, both w.r.t reliability (see "The (un)reliability of saliency maps") and added insight -- in the qualitative eg shown, I don't think the saliency map tells us anything new.

Related work is limited on the technical side. Adding works for the 'generalized' approach would be important (see Somepalli's CSD work as a good starting point), and it is worth noting that others have proposed more or less the same approach as this paper previously: see Casper et al's "Measuring the success of diffusion models at imitating human artists" and Moayeri et al's "Rethinking Artistic copyright ..." -- in fairness, these are workshop papers, so I don't think they detract from the novelty of this paper, but these could still be worth looking over / citing.


Minor typos / nitpicks:
- L85: x_j not defined -- perhaps the right statement is "1 if there exists x_j s.t. f(q_i, x_j) > alpha" (there exists a training sample that is highly similar to one instance in the query/plaintiff set"
- L183: parentheses not needed when using citep -- result is double parantheses
- Figure 1: (a) who is Rita? Did you mean Mary? (b) save your figure with dpi=300 or as a pdf to increase the resolution
- L215: "a cartoons"
- L225 : "demonstraes"
- Some missing citations: L241, court cases in L150
- L381 extra space / incomplete sentence -- "as the defendant set" perhaps?

**Questions:**

How should a legal audience interpret a similarity score of 0.687? I appreciate that you explicitly state that this analysis should not stand alone, but even with this said, it is unclear how a given score should be interpreted / how to go from a score to a judgement on infringement.

How should one select 'distractor' classes? It seems as though outputs of the method are quite sensitive to this choice.

What is the purpose of the template matching in fig 5? This part was not explained.

Can you provide more detailed comparisons to existing approaches? The contribution is not well situated in prior work.

**Details Of Ethics Concerns:**

n/a -- in fact, I really appreciate the intentional effort to recognize the sociotechnical nature of this problem and consider its implications on the *people* affected.

---

### Note · Authors · 2024-11-18

I have read and agree with the venue's withdrawal policy on behalf of myself and my co-authors.